# Computerized neuropsychological assessment in post-COVID condition

**Cristina Delgado-Alonso[1], Jordi A. Matias-Guiu ID[1]\*, Jesús M. Alvarado ID[2],
Maria Diez-Cirarda[1], Silvia Oliver-Mas[1], Esther Valiente-Gordillo[1],
María José Gil-Moreno[1], José-Manuel Alcalá Ramírez del Puerto[1],
Jorge Matías-Guiu[1], Alfonso Delgado-Álvarez ID[2]**

1 Department of Neurology, Hospital Clínico San Carlos. San Carlos Health Research Institute (IdISSC), Universidad Complutense de Madrid, Madrid, Spain, 2 Department of Psychobiology & Behavioral Sciences Methods, Universidad Complutense de Madrid, Madrid, Spain

\* jordimatiasguiu@hotmail.com, jordi.matias-guiu@salud.madrid.org

## Abstract

### Background

Attention/ processing speed deficits with or without executive function and episodic memory deficits have been suggested as a relatively characteristic cognitive profile of people with post-COVID condition (PCC). Most studies have been performed using standardized paper and pencil neuropsychological assessment. Sensitive and applicable tests are needed to improve the diagnostic capacity of patients with PCC.

### Objectives

In this study, we aimed to investigate the dimensions of a comprehensive computerized neuropsychological battery and to characterize the cognitive characteristics of patients with PCC.

### Materials and methods

Five hundred and eight participants were enrolled in the study (PCC = 227, Healthy Controls, HC = 281) and underwent cognitive assessment focused on attention, concentration, executive functions, and episodic memory. We conducted a multi-group confirmatory factor analysis. Factor scores were obtained to compare the PCC and HC groups and partial invariance analysis was performed to identify relevant cognitive processes that differentiate the two groups.

### Results

The proposed four-factor model showed adequate fit indices. There were differences in attention, concentration, and executive functions factor scores with small to moderate effect sizes and with a particular implication of attention processes based on measurement invariance analysis. Impairments in reaction times and divided attention were especially relevant in patients with PCC.

**Data availability statement:** The conditions of our ethics approval do not permit public archiving of anonymized study data. Readers seeking access to the data should contact the corresponding or the local ethics committee of Hospital Clinico San Carlos, Madrid (E-mail: ceic.hcsc@salud.madrid.org). Access can be granted only to named individuals in accordance with ethical procedures governing the reuse of sensitive clinical data.

**Funding:** The study was partially supported by Schuhfried though the project 20/022-E of normative data collection. This research has also received funding from the Nominative Grant FIBHCSC 2020 COVID-19 (Department of Health, Community of Madrid). Jordi A. Matias-Guiu is supported by Instituto de Salud Carlos III through the project INT20/00079 and INT23/00017 (co-funded by European Union). Maria Diez-Cirarda is supported by Instituto de Salud Carlos III (ISCIII) through Sara Borrell postdoctoral fellowship Grant CD22/00043) and co-funded by the European Union. Silvia Oliver-Mas is supported by the Fundación para el Conocimiento madri+d through project G63-HEALTHSTARPLUS-HSP4. The sponsors of the study did not take part in the design and conduct of the study; collection, management, analysis, and interpretation of the data; writing and review of the report; or the decision to submit the article for publication.

**Competing interests:** The authors have declared that no competing interests exist.

## Conclusions

The battery revealed four factors representing attention, concentration, executive functions, and episodic memory. The PCC group performed worse than the HC group in attention, concentration, and executive functions. These findings suggest the validity of computerized neuropsychological assessment, which could be particularly useful in PCC.

## Introduction

Post-COVID condition (PCC) has been defined as the development of symptoms during or after the infection of SARS-CoV-2 that persist over 12 weeks [1]. PCC symptomatology consists of a wide range of symptoms with a relevant impact on functioning, quality of life, and important socio-economic consequences [2]. According to previous meta-analyses, some of the most reported symptoms are chronic fatigue, dyspnea, pain, depression, anxiety, headache, insomnia, smell or taste disorders, and subjective cognitive complaints (e.g., brain fog, memory issues) [3,4].

Objective cognitive impairment has been previously shown and include mild to moderate deficits in attention, executive function, episodic memory, visuospatial function, and language [5–8]. Some studies have suggested a relatively characteristic cognitive profile of PCC, consisting of attention/processing speed deficits with or without episodic memory and executive function impairment [9,10].

The cognitive assessment of people with PCC is key to defining an individual cognitive profile that guides cognitive training/ rehabilitation, supports enrollment in clinical trials, and justifies possible job adaptations and/ or disability benefits [9].

Paper-and-pencil tests have been most commonly used to assess the cognitive symptoms of PCC [11]. Different strategies have been adopted for the cognitive assessment of people with PCC, including screening tests (e.g., Montreal Cognitive Assessment, MoCA), telephone cognitive evaluations (e.g., Telephone Interview for Cognitive Status, TICS), or internet-based assessments (e.g., iPad-based online neuropsychological tests) [8,12–14]. However, cognitive screening tests do not allow an in-depth description of the cognitive processes underlying cognitive impairment, and dementia screening tests might not be suitable for people with PCC, whose deficits require tests with more sensitivity than people with dementia. In addition, conducting a cognitive assessment in uncontrolled settings could bias the results obtained. Similar to other conditions, remote assessments (telephone or internet) may include factors such as variability in the environment affecting performance or lack of direct supervision. The use of computerized tasks when cognitive impairment is suspected should be considered for their impact on cognitive performance and the presence of an examiner may be necessary [15,16]. For these reasons, a comprehensive neuropsychological battery conducted in-person and in a controlled and structured setting seems more recommended.

While paper and pencil-based assessments are frequently administered during a comprehensive neuropsychological evaluation, computerized assessments could

have important advantages. First, computerized assessments are particularly useful for time-dependent variables that cannot be easily and reliably collected by traditional paper and pencil-based evaluations (e.g., reaction times). Thus, computerized tests may have more sensitivity for those cognitive processes where time measures play a significant role, such as attention processes [17]. Second, the administration is automated. Although a supervisor is always highly recommended during the assessment, it is possible to conduct more than one assessment at the same time. Third, they facilitate real-time scoring, minimizing errors, and generating reports immediately for faster clinical decision-making. Fourth, as a standardized protocol, it reduces intra- and inter-rater variability. Finally, they offer different parallel versions to prevent practice effects in case of repeated measures, improving accuracy and validity [18]. Moreover, computerized tools can also be useful in the rehabilitation of cognitive deficits in various neurological disorders. Online interventions have been shown to be useful in the treatment of prolonged cognitive impairment (e.g., Long COVID) [19]. Despite these advantages, it is necessary to analyze the validity of these batteries assessing their dimensionality and factorial structure, because the differences in the administration procedure could produce different outcomes.

In the context of PCC, computerized tests have been shown to be useful for evaluating patients on a large scale [20], and also may be more sensitive than classic paper-and-pencil tasks and appropriate to assess specific domains such as attentional o processing speed [17,21].

An inter-group comparison between cognitively healthy control (HC) and PCC groups in a test set or neuropsychological battery is commonly used for the detection of cognitive deficits. However, this can be discouraged when several scores are available for comparison, increasing type I error or applying corrections for multiple comparisons, for instance, Bonferroni correction, which tends to be very conservative when many comparisons have been made [22]. In this regard, confirmatory factor analysis (CFA) could be especially helpful, as a technique that condenses several test scores into a smaller number of factors (also called latent variables) following the theoretical framework that suggests the way test scores are grouped. CFA can explore the latent structure of a neuropsychological battery by describing the number of factors and the pattern of test-factor relationship under certain conditions [23,24]. For instance, how scores in divided, alert, and selective attention tests are related to the factor "attention". Thus, CFA is understood as a source of validity evidence based on internal structure [25].

In this study, we conducted a computerized neuropsychological assessment based on a comprehensive neuropsychological assessment administered using the Vienna Test System (VTS) (Schuhfried GmbH; Moedling, Austria) [26] in HC and PCC groups, covering the cognitive domains characteristic of the PCC profile.

The primary aim was to explore the factor structure of VTS in a combined sample of PCC and HC, and the secondary aim was examine specific deficits in PCC. We aimed (i) to conduct a multi-group CFA, (ii) to compare HC and PCC groups in their factor scores, (iii) to describe the cognitive processes underlying measurement invariance based on factorial invariance analysis.

We hypothesize that a four-factor model could show adequate fit indices and be shared by both groups with differences in their factor scores. A failure to find metric and scalar invariance is expected, implying measures of impaired cognitive processes in PCC.

## Methods

### Participants

Five hundred and eight participants, whose first language was Spanish, were recruited from the Department of Neurology at Hospital Clinico San Carlos. The total sample consisted of cognitively healthy control participants (HC) (n = 281) and participants with post-COVID condition (PCC) (n = 227). The mean time from COVID-19 onset to assessment was 17.59 ± 8.86 months.

Inclusion criteria for participants with PCC were as follows: (1) a diagnosis of COVID-19 (positive RT-PCR) at least 6 months before inclusion in the study, and (2) a diagnosis of PCC according to the World Health Organization criteria [1],

and (3) cognitive complaints related with the SARS-CoV-2 infection. The HC group met the following criteria: (1) Age, sex, and educational level according to the study population, (2) absence of cognitive impairment (CDR score of 0) [27], and (3) absence of functional impairment based on scores of 0 in the Functional Activities Questionnaire [28]. The exclusion criteria for all participants were: (1) currently no medical treatment for or no diagnosis of any neurological, psychiatric, or medical disorder with potential impact on cognition (e.g., epilepsy, major depression), (2) prior history of substance abuse (e.g., alcohol), and (3) any physical difficulty leading to potential bias in scores (e.g., hearing deficits).

All patients included in the study met the WHO criteria for PCC [1]. For patient selection, both physical and neurological examinations were performed. Patients with any neurological disorder involving cognitive impairment (including mild cognitive impairment) or other systemic or psychiatric disorders potentially impacting the findings were excluded. All patients underwent MRI to exclude other causes, and additional tests in selected cases (e.g., CSF biomarkers or FDG-PET) were performed based on clinical suspicion. Information on the clinical characteristics of the disease was cross-checked with the hospital and regional medical chart (e.g., hospital admission or ICU admission). Patients were selected with a 6-month criterion to separate possible symptoms that may occur after the acute or subacute phase of the disease (inflammation) and persistent symptoms. This criterion was established to be able to separate those possible cases where there may be a spontaneous recovery and to recognize those patients in whom the symptoms have become chronic [29].

### Neuropsychological measures

All participants completed the computerized neuropsychological battery Vienna Test System (VTS) (GmbH; Schuhfried & Moedling, Austria) [26] assess attention, concentration, executive functions, and episodic memory using the following tests: the Figural Memory Test (FGT) (S11 form), the Tower of London (TOL-F) (S1 form), the Trail Making Test (TMT) (S1 form), a variant of the go/no-go task (INHIB) (S13 form), and the N-back verbal task (NBV) (S1 form), as part of the Cognitive Basic Testing (COGBAT) battery [30]. Additionally, both groups were assessed using the Reaction Test (RT) (S3 form) [31] and the Perception and Attention Function Battery (WAF) (S1 form) [32] to evaluate specific attention processes, such as alertness, vigilance and sustained attention, divided attention, selective attention, smooth pursuit eye movements, and spatial attention. To assess concentration, the Cognitrone test (COG) (S11 form) [33] and the Determination Test (DT) (S1 form) [34] were administered. The computerized battery was administered in person at our hospital by trained neuropsychologists. It was conducted in two-day sessions. Each session lasted approximately 80 minutes. All measures for each test are shown in Table 1. Furthermore, we administered the Beck Depression Inventory (BDI) [35] and the Pittsburgh Sleep Quality Index (PSQI) [36], to evaluate depressive symptoms and sleep quality. Additionally, the Modified Fatigue Impact Scale (MFIS) [37] was administered to assess fatigue at the time of assessment. The assessment evaluates the impact of fatigue in the past 4 weeks.

We distinguished between *'objective cognitive impairment,'* which refers to impairment confirmed through neuropsychological assessment, and *'subjective cognitive complaints',* which refer to self-reported cognitive issues irrespective of test results.

### Procedure

A cross-sectional study was conducted with the approval of the Ethics Committee of Hospital Clinico San Carlos and all participants provided written informed consent. Participants were consecutively recruited through the Department of Neurology at the Hospital Clinico San Carlos between 1th November 2020 and 31th July 2022.

The cognitive assessment was conducted in Spanish by trained neurolopsychologists in two independent sessions lasting approximately 80 minutes each.

We conducted a multi-group confirmatory factor analysis (CFA) considering the main cognitive variables of the VTS battery to study the validity evidence based on internal structure. Factor scores were obtained to compare the PCC and

**Table 1. Cognitive variables of the Vienna Test System battery considered in the multi-group CFA.**

| FACTOR | TEST | TIME | MEASURES | ABBREVIATION |
|---|---|---|---|---|
| **Attention Processes** | Perception and | 17 min | (1) Alertness – mean reaction time | WAFalert |
| | Attention | | (2) Divided attention – mean reaction time | WAFdivid |
| | Function Battery | | (3) Selective attention – mean reaction time | WAFselect |
| | | | (4) Neglect – mean reaction time | WAFnegl |
| | | | (5) Spatial attention – mean reaction time | WAFspace |
| | | | (6) Smooth pursuit eye movements – mean reaction time | WAFeye |
| | Reaction Test | 6 min | (1) Motor speed | RTmotor |
| | | | (2) Reaction speed | RTreaction |
| **Concentration** | Cognitrone | 10 min | (1) Mean time correct rejections | COGtime |
| | | | (2) Total correct rejections | COGcorrect |
| | Determination Time | 6 min | (1) Accurate reactions | DT |
| **Executive Function** | Trail Making Test | – | (1) Working time part A | TMTa |
| | | | (2) Working time part B | TMTb |
| | Tower of London | 16 min | (1) Planning ability | TOL |
| | N-Back verbal | 9 min | (1) Total errors | NBVerrors |
| | | | (2) Total correct responses | NBVcorrect |
| | Go/no-go | 1 min | (1) Number of commission errors | INHIB |
| **Memory** | Figural Memory Test | 14 min + 35 min delay | (1) Learning total | FGTlearning |
| | | | (2) Short-term free recall (5 min) | FGTshort |
| | | | (3) Long-term free recall (30 min) | FGTlong |
| | | | (4) Correct recognition | FGTrecog |

HC groups (Fig 1) and partial invariance analysis was performed to identify relevant cognitive processes that differentiate the two groups.

Different groups may be considered in the same CFA. Under this condition, a multi-group CFA is conducted to check if the neuropsychological tests represent the same theoretical constructs between groups (*measurement invariance*) [38]. The different levels of measurement invariance can contribute to a better understanding of the cognitive processes underlying PCC. In this regard, HC and PCC groups can share the same latent structure of a particular neuropsychological battery (*configural invariance*). But not necessarily all the tests have the same relationship strength with their factors (*metric invariance* or *weak invariance*). For instance, the strength of the relationship between specific measures of attention and the attention factor may vary between groups. At the same level of attention factor, both HC and PCC groups could not be at the same level of each administered attention test (*scalar invariance* or *strong invariance*). Other invariance levels could be of interest but overly restrictive, such as residual invariance [38]. A failure to find a certain level of invariance and the identification of responsible test scores can help to understand the cognitive profile behind PCC. Additionally, this approach is especially relevant considering that there is not a gold standard for the neuropsychological diagnosis of PCC.

## Determination of the most likely SARS-CoV-2 variants and vaccination status

Based on the first date of symptom, onset of the acute infection was used to determine the waves of infection: one hundred forty patients (61.7%) belonged to the first wave (March-May 2020), thirty-eight (16.7%) to the second wave (alpha variant, August-November 2020), seventeen (7.5%) to the third (delta, December 2020-July 2021), and thirty-two (14%) to the fourth or subsequent waves. Data from the Spanish Ministry of Health, 2024 [39].

Regarding the vaccination status: 221 (96.4%) patients were not vaccinated on the first date of infection. At the time of cognitive assessment, 156 (68.7%) of patients were vaccinated.

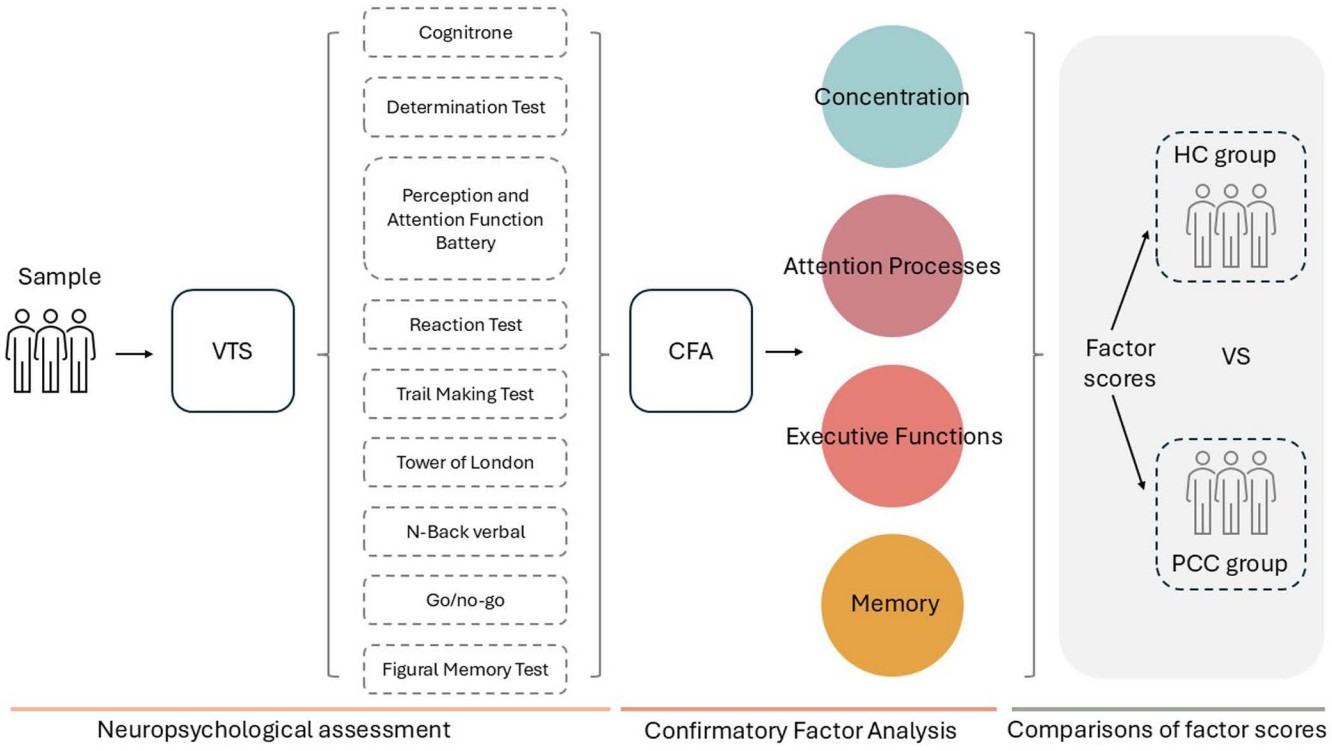

**Fig 1. Summary of the procedure for obtaining factor scores and performing comparisons between groups.**

## Statistical analysis

Statistical analysis was performed using RStudio 4.3.1, with alpha set at 0.05. Descriptive data are presented as mean ± standard deviation or frequencies.

For comparisons between PCC and HC groups, the t-tests were performed, and Pearson's chi-squared test was used to assess the independence of categorical variables. Spearman's correlation was used to measure relationships between quantitative variables and was categorized as very low (0-.29), low (.30-.49), moderate (.50-.69), high (.70-.89), or very high (>.89).

The CFA was performed using the "*lavaan*" package [40]. We considered four latent factors and robust maximum likelihood estimation. Model fit was evaluated based on the following thresholds: Comparative Fit Index (CFI), Tucker-Lewis index (TLI) ≥.0.95, standardized root mean squared residual (SRMR) ≤ 0.08, and root mean square error of approximation (RMSEA) ≤ 0.06 [41]McDonald's omega ($\omega$) was reported as a reliability estimator.

A two-way mixed ANOVA was calculated to compare HC and PCC, considering factor scores as the within-subjects variable and group as the between-subjects variable. We checked the multi-sample sphericity assumption using Mauchly's sphericity test and Box's test, and the homogeneity of variance assumption was checked using Levene's test. In the case of not assuming sphericity, the multivariate F test was used. Post hoc comparisons were adjusted using Bonferroni's correction for multiple comparisons. Partial eta squared was used as a measure of effect size, considered as small (partial $\eta^2 = 0.01$), medium (partial $\eta^2 = 0.06$), and large (partial $\eta^2 = 0.14$).

We conducted a partial factorial invariance analysis to investigate the role of specific cognitive processes, based on VTS scores, in measurement invariance. Particularly, we focused on configural, metric (weak), and scalar (strong) invariance. Chi-squared tests and the difference in the comparative fit index (CFI) (cutoff point ΔCFI < .01) were used to

determine substantial increases in fit [42]. The proposed factorial model was fitted independently for PCC and HC to study configural invariance, reporting similar model fit indices. The identification of potential invariance scores was performed by inspecting the modification indices for metric and scalar invariance.

## Results

### Characteristics of the sample

The main demographic and clinical characteristics are described in Table 2. There were no significant differences between PCC and HC groups in age ($t = 0.25$, $p = .80$) or years of education ($t = 1.40$, $p = .16$). The percentage of females was higher in the PCC group (78% vs 56.9%, $\chi^2(1) = 24.3$, $p < .001$) (Table 2).

### Confirmatory factor analysis

A CFA was conducted based on the main cognitive variables of the computerized neuropsychological battery VTS (Fig 2). The four-factors model showed adequate fit measures: $\chi^2(183) = 362.32$, $p < .001$; CFI = 0.95, TLI = 0.94, SRMR = 0.05, and RMSEA = 0.044 (IC: 0.039–0.049). McDonald's omega was 0.92 for memory, 0.87 for attention processes, 0.70 for executive functions, and 0.65 for concentration.

### Comparisons between PCC and HC

Factor scores were obtained to compare PCC and HC groups in attention, concentration, executive functions, and episodic memory. We found a significant interaction effect (group x factor scores) (multivariate $F (3,504) = 16.96$, $p < .001$,

**Table 2. Main demographic and clinical characteristics of the sample.**

|  | pwPCC | HC |
|---|---|---|
| **n** | 227 | 281 |
| **Age (years)** | 45.57 ± 10.31 | 47.23 ± 18 |
| **Sex (% female)** | 78% | 56.9% |
| **Education (years)** | 15.2 ± 3.28 | 14.79 ± 3.26 |
| **Evolution (months)** | 17.59 ± 8.86 |  |
| **Number of infections** |  |  |
| Two | 21.1% |  |
| Three | 3.5% |  |
| **Premorbid Risk Factors** |  |  |
| Dyslipidemia | 25.2% |  |
| Hypertension | 16.38% |  |
| Smoking habit | 14.6% |  |
| Coronary artery disease | 8.4% |  |
| Diabetes | 8.4% |  |
| **Medical care in acute stage** |  |  |
| Hospital admission | 23.9% (54/226) |  |
| ICU admission | 4.9% (11/214) |  |
| **Symptoms in acute stage** |  |  |
| Headache | 81% |  |
| Hyposmia & Ageusia | 58.38% |  |
| Confusion | 50% |  |

pwPCC: people with Post-Covid Condition; HC: cognitively healthy control participants.

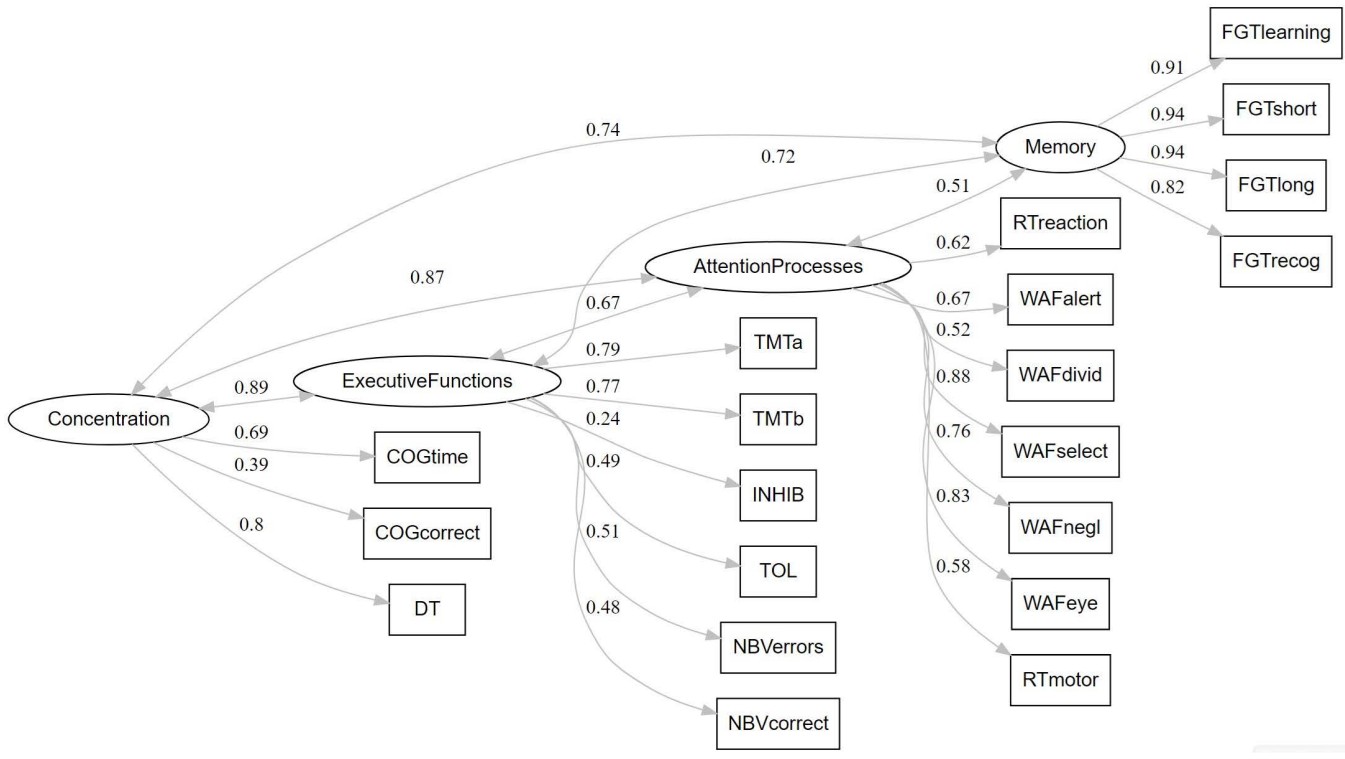

**Fig 2. Representation of the four-factors model with all significant standardized values (factor loadings and factor covariances as absolute values).**

partial $\eta^2 = .092$). Post-hoc comparisons showed a worse performance by PCC group compared to HC in Attention Processes ($F(1,506) = 37.29$, $p < .001$, partial $\eta^2 = .069$), Concentration ($F(1,506) = 14.58$, $p < .001$, partial $\eta^2 = .028$), and Executive Function ($F(1,506) = 5.6$, $p < .018$, partial $\eta^2 = .011$), but not in the Memory score ($F(1,506) = 0.29$, $p = .59$, partial $\eta^2 = .001$).

Correlations between factor scores and clinical characteristics of the PCC sample are shown in Fig 3. There were no interaction effects in hospital admission (hospital admission x factor scores) (multivariate $F(3,222) = 0.70$, $p < .555$, partial $\eta^2 = .009$) or a main effect ($F(1,224) = 0.99$, $p < .32$, partial $\eta^2 = .004$). Similarly, we did not find a significant interaction effect in ICU admission (ICU admission x factor scores) (multivariate $F(3,210) = 0.12$, $p < .94$, partial $\eta^2 = .002$) or a main effect ($F(1,212) = 0.043$, $p < .83$, partial $\eta^2 < .001$).

### Cognitive processes underlying measurement invariance

The proposed model was fitted for HC and PCC groups independently. Both groups showed similar model fit indices, considering the reduction in the sample size. In the HC group, $\chi^2(183) = 320.41$, $p < .001$; CFI $= 0.95$, TLI $= 0.94$, SRMR $= 0.05$, and RMSEA $= 0.052$ (IC: 0.043–0.060). In the PCC group, $\chi^2(183) = 277.64$, $p < .001$; CFI $= 0.93$, TLI $= 0.92$, SRMR $= 0.07$, and RMSEA $= 0.048$ (IC: 0.039–0.056).

The results of the measurement invariance and partial measurement invariance are shown in Table 3. We found significant contributions of RT-motor scores in metric invariance (loadings) and WAF-divided and RT-reaction scores in scalar invariance (intercepts).

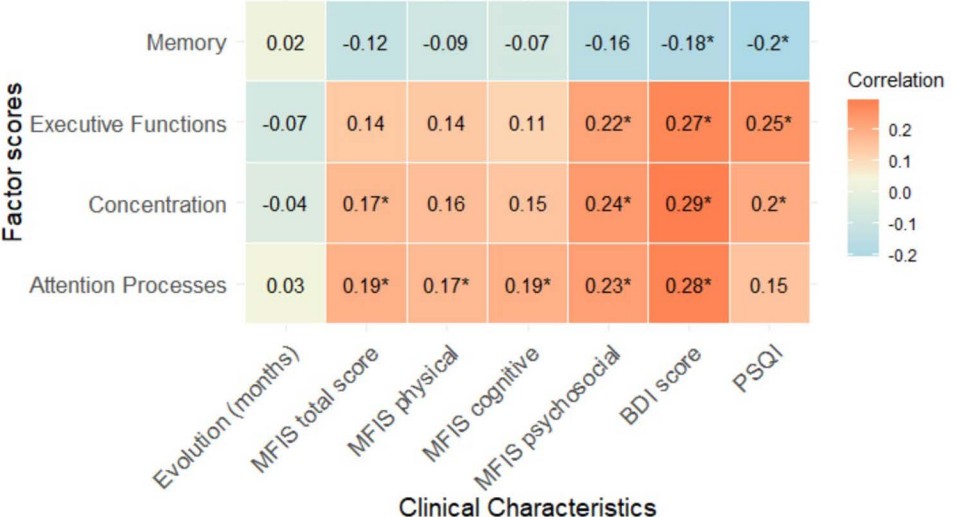

**Fig 3. Heatmap showing correlations (rho) between clinical characteristics of PCC groups and factor scores.** MFIS: Modified Fatigue Impact Scale; BDI: Beck Depression Inventory; PSQI: Pittsburgh Sleep Quality Index. *: significant correlations after Bonferroni's correction.

**Table 3. Series of model comparisons.**

| Level of invariance | ΔX2 | ΔDf | Δp | ΔCFI |
|---|---|---|---|---|
| **M1 configural** | | | | |
| **M2 metric** | 35.48 | 16 | 0.003* | 0.009 |
| **M2 metric** # except RT-motor | 23.04 | 15 | 0.083 | 0.004 |
| **M3 scalar** | 51.88 | 15 | 5.91e-06* | 0.005 |
| **M3 scalar** # except WAD-divided | 29.92 | 14 | 0.008* | 0.002 |
| **M3 scalar** # except WAD-divided+RT-reaction | 21.21 | 13 | 0.069 | 0.001 |
| **M4 strict** | 29.88 | 19 | 0.053 | 0.014· |
| **M5 equal means** | 185.11 | 4 | 2e-16* | 0.008 |

M: model, Δ: increase, DF: degrees of freedom.

*: significant chi-squared test.

·: ΔCFI > 0.01.

## Discussion

In the present study, we aimed to explore the internal structure of a computerized neuropsychological battery administered using the VTS in participants with PCC and HC. Based on the composition of the computerized battery, we proposed a four-factor model encompassing attention processes, concentration, executive functions, and memory, which demonstrated adequate fit indices.

The four-factor model aligns with the cognitive processes assessed by the computerized battery. The Perception and Attention Function Battery (WAF) and the Reaction Test (RT) contribute notably, facilitating the evaluation of different

attentional processes and reaction times that are challenging to measure with traditional paper and pencil-based assessments. The Cognitrone (COG) test and the Determination Test (DT) test were specially designed to assess concentration as a single dimension closely related to attention [33,34]. Executive function, a broad domain encompassing high-level processes essential for adapting to novel or complex situations [43], is assessed through several tests, including the Trail Making Test (TMT), the Tower of London (TOL-F), a variant of the go/ no-go task (INHIB), and the N-Back Verbal task (NBV). These tests cover a range of cognitive processes, such as mental flexibility, planning, inhibition, and working memory. Finally, the Figural Memory Test (FGT) was designed to assess episodic memory and its observed variables showed the highest loadings in the model. As expected, the four factors were highly correlated, reflecting the interrelatedness of the cognitive domains assessed.

The computerized battery showed evidence of partial strong invariance after three adjustments. In addition, the study of measurement invariance revealed specific cognitive impairments in the PCC group. Both PCC and HC groups shared the same factor structure, supporting configural invariance. The strength of relationships between tests and factors was the same across groups, except for RT-motor scores, suggesting partial metric invariance and different loadings between RT-motor scores and the factor Attention Processes depending on whether the condition was PCC or HC. With the exception of WAF-divided and RT-reaction scores, the VTS battery showed partial scalar invariance when the model constraint for equivalent test intercepts across groups for all indicators. WAF-divided and RT-reaction indicators were linked to the Attention Processes factor similarly in both PCC and HC groups. However, intercepts differed, and observed scores would vary at different levels of Attention Processes [38].

Significant differences were found between PCC and HC groups in Attention Processes, Concentration, and Executive Functions with worse performance in the PCC group. The effect sizes were medium for Attention Processes and small-to-medium for Concentration and Executive Functions. Interestingly, no significant difference was found in Memory scores. Previous studies have highlighted attentional deficits in PCC [4,5] and have suggested attentional deficits as a characteristic feature of PCC [9,10] when other cognitive domains, such as memory, executive function, language, and visuospatial skills are also studied. Attention has been previously reported as the most frequent cognitive domain impaired, followed by episodic memory, executive function, visuospatial function, and language. In our study, divided attention and reaction time measures had important implications in the study of measurement invariance. The differences observed in divided attention and reaction time measures are in line with the commonly reported "*brain frog*" and a decrement in reaction time tasks has been described previously in PCC [44]. Interestingly, cognitive slowing and difficulties in multitasking are common symptoms reported by patients with PCC [45]. In contrast to some studies, we did not find any difference in episodic memory [4,5]. This finding could be explained because the decrements in episodic memory in PCC are generally lower than in attention, and we included a visual memory task, which could be less sensitive than verbal episodic memory tests. Recent studies have shown that visual memory may be less sensitive for detect cognitive deficits compared to the more prevalent verbal memory impairment [46,47].

Another important finding is that the factor scores showed low, mostly nonsignificant correlations with other non-cognitive characteristics of the PCC sample (e.g., depression, anxiety, fatigue). The influence of neuropsychiatric symptoms in cognitive performance in PCC is still a controversial issue in the literature [6,9]. Future studies comparing different cognitive tasks and procedures of administration could be of interest to evaluate whether computerized assessments could show different vulnerabilities to the effects of anxiety or depression in this setting compared with the standard cognitive assessment conducted by a neuropsychologist.

Our study has some limitations that should be considered. First, both PCC and HC groups had an unequal sex distribution, with more females in the PCC groups, reflecting the prevalence of PCC [48]. Second, we did not consider other cognitive domains, such as language or visuospatial skills. However, these domains seem to have a less prominent role in PCC cognitive dysfunction. Third, data on antiretroviral treatment during the acute phase were not included because they were not available. However, in our sample, only 23.9% of patients were hospitalized. Thus, we believe that this should

not impact on the results because these therapies were mainly used in hospitalized patients. Fourth, we did not include performance validity tests [49], which could be of interest in future studies.

In conclusion, the internal structure of the computerized battery revealed four factors representing attention, concentration, executive functions, and episodic memory. The PCC group performed worse than the HC group in attention, concentration, and executive function. Attentional deficits, particularly in divided attention and reaction times, were prominent findings, suggesting that these specific cognitive deficits could be hallmarks of cognitive dysfunction in PCC. Overall, our study suggests the validity of computerized neuropsychological assessment, which could be particularly useful in PCC for diagnosis, monitoring, and clinical trials.

## Author contributions

**Conceptualization:** Jordi A Matias-Guiu, Jorge Matias-Guiu, Alfonso Delgado-Álvarez.

**Data curation:** Cristina Delgado-Alonso, Jordi A Matias-Guiu, Maria Diez-Cirarda, Silvia Oliver-Mas, Esther Valiente-Gordillo, María José Gil-Moreno, José-Manuel Alcalá Ramírez del Puerto, Alfonso Delgado-Álvarez.

**Formal analysis:** Cristina Delgado-Alonso, Jordi A Matias-Guiu, Jesús M Alvarado, Alfonso Delgado-Álvarez.

**Funding acquisition:** Jordi A Matias-Guiu, Jorge Matias-Guiu.

**Investigation:** Cristina Delgado-Alonso, Jordi A Matias-Guiu, Maria Diez-Cirarda, Silvia Oliver-Mas, Esther Valiente-Gordillo, María José Gil-Moreno, José-Manuel Alcalá Ramírez del Puerto, Jorge Matias-Guiu, Alfonso Delgado-Álvarez.

**Methodology:** Jordi A Matias-Guiu, Jesús M Alvarado, Alfonso Delgado-Álvarez.

**Supervision:** Jordi A Matias-Guiu, Alfonso Delgado-Álvarez.

**Visualization:** Jordi A Matias-Guiu, Jorge Matias-Guiu, Alfonso Delgado-Álvarez.

**Writing – original draft:** Cristina Delgado-Alonso, Jordi A Matias-Guiu, Alfonso Delgado-Álvarez.

**Writing – review & editing:** Cristina Delgado-Alonso, Jordi A Matias-Guiu, Jesús M Alvarado, Maria Diez-Cirarda, Silvia Oliver-Mas, Esther Valiente-Gordillo, María José Gil-Moreno, José-Manuel Alcalá Ramírez del Puerto, Jorge Matias-Guiu, Alfonso Delgado-Álvarez.

## Acknowledgments

We thanks all the patients with PCC involved in the study.

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
