## [Decision Letter · Decision Letter 0]

5 Dec 2024

PONE-D-24-46762Computerized neuropsychological assessment in post-COVID conditionPLOS ONE

Dear Dr. Matias-Guiu,

Thank you for submitting your manuscript to PLOS ONE. After careful consideration, we feel that it has merit but does not fully meet PLOS ONE’s publication criteria as it currently stands. Therefore, we invite you to submit a revised version of the manuscript that addresses the points raised during the review process.

Thank you for addressing the reviewers' comments in a revised version while focusing on reviewer 3 and reviewer 1 comments.

In addition, please provide greater details regarding how PCC (WHO) definition was exactly applied in the PCC sample. That is for each individual how the WHO was applied on each relevant criterion of the definition and using which tools.

Was fatigue measured at any stage? And other key post-viral syndromes? Was this also completed in the health controls? How comorbid causes were excluded (or not) as explanatory (or not) of the current post-viral symptoms as required by the WHO definition? Was a clinical examination conducted and if so by whom and using which instruments?

This additional information is needed to establish the robustness of the PCC definition in the current study.

Furthermore, please provide additional information regarding acute infection including, how long after acute COVID infection, were the participants evaluated? What types of SARS-CoV-2 variants were most prevalent at the time of this study (by participants of these data is available), how many were vaccinated, how many have received antiretrovirals? Include these as limitations if these data were not available and discuss how this may have impacted the results.

Could you also explain why the current study participants were recruited from a Neurology tertiary service? What are the consequences of the current samples’ ascertainment?

These data are needed to better establish the representativeness of the sample in comparison to the Spanish general population at the same time of the pandemic. Please include references to relevant Spanish general population data.

We look forward to receiving your revised manuscript.

Kind regards,

Lucette A Cysique, PhD

Academic Editor

PLOS ONE

2. In this instance it seems there may be acceptable restrictions in place that prevent the public sharing of your minimal data. However, in line with our goal of ensuring long-term data availability to all interested researchers, PLOS’ Data Policy states that authors cannot be the sole named individuals responsible for ensuring data access (http://journals.plos.org/plosone/s/data-availability#loc-acceptable-data-sharing-methods).

Reviewers' comments:

Reviewer's Responses to Questions

**Comments to the Author**

1. Is the manuscript technically sound, and do the data support the conclusions?

Reviewer #1: Yes

Reviewer #2: Partly

Reviewer #3: Yes

2. Has the statistical analysis been performed appropriately and rigorously? 

Reviewer #1: Yes

Reviewer #2: Yes

Reviewer #3: I Don't Know

3. Have the authors made all data underlying the findings in their manuscript fully available?

Reviewer #1: Yes

Reviewer #2: Yes

Reviewer #3: No

4. Is the manuscript presented in an intelligible fashion and written in standard English?

Reviewer #1: Yes

Reviewer #2: Yes

Reviewer #3: Yes

5. Review Comments to the Author

Reviewer #1: Thank you for allowing me to review this interesting and clinically-needed research.

Throughout the paper, the term Post COVID condition (PCC) is used. It is unclear if this refers only to persistent cognitive symptoms. Please include how you are defining the term.

Can you please describe why 6 months was chosen for persistent symptoms, as opposed to 3 months which is the definition of PCC.

Could the HC group include other forms of Long COVID? For instance, for people who report post-covid persistent changes in smell perception, what group would they go into?

Methods- Can more detail be provided on the administration of the measures. Were they done in person or remotely? What was the role of the administering neurologists? Did they read instructions and clarify?

Can discussion of whether the battery could be used remotely be added?

Has the Vienna Test System been validated? Has it been shown to correlate with other neuropsycological measures?

Edits for clarification:

Page 4 last paragraph:

But not necessarily all the tests have the same relationship strength with their factors (metric invariance or weak invariance).

Reviewer #2: Dear Sir/Mam

Please find bellow the requested review regarding the manuscript. The article contains a lot of useful information on the issue. The topic is very interesting and use of sources is appropriate. Although it has some useful information there are less references and the statements are not established. I suggest the authors to write more information with references.

The article contains a lot of useful information on the issue. It is quite clear what is already known about this topic and the research question is clearly outlined. The abstract is good and introduction section involves too much information. The introduction should be briefer and the authors should include the studies of:

• Megari, Kalliopi, Evanthia Thomaidou and Electra Chatzidimitriou (2024). Highlighting the Neuropsychological Consequences of COVID-19: Evidence from a Narrative Review. INQUIRY: The Journal of Health Care Organization, Provision, and Financing, Volume 61: 1 –10. DOI: 10.1177/00469580241262442

• Akyllina Despoti, Kalliopi Megari, Anna Tsiakiri, Maida Toumaian, Vasiliki Koutzmpi, Athanasia Liozidou, Angeliki Tsapanou. Effectiveness of remote neuropsychological interventions: a Systematic Review. Applied Neuropsychology Adult 27:1-9. doi: 10.1080/23279095.2024.2382814

The research question is justified clearly, given what is already known about the topic. The results are not discussed from multiple angles and conclusions answer the aims of the study partially. The conclusions are partially supported by references or results and the limitations of the study fatal and it is questionable if there are opportunities to inform future research. Positive: There are some strengths of the article that could have an impact in the field, such as the topic and its impact on the existed literature.

Reviewer #3: Introduction:

- Page 3, paragraph 1 and 2 refers to “cognitive symptoms” is unclear whether this is self-report or performance based. It would be helpful to have them labeled more specifically (e.g., self-reported cognitive symptoms and Neuropsychological test performance/impairments on neuropsychological tests) for the introduction and also discussion.

- I think it would be helpful to add that the majority of post COVID studies have been using pencil and paper comprehensive evaluations. Since these are not mentioned until paragraph 4, it seemed like these were used less often than the other approaches mentioned in paragraph 3.

- “However, cognitive screening tests do not allow an in-depth description of the cognitive processes underlying cognitive impairment, and dementia screening tests might not be suitable for people with PCC, whose deficits require tests with more sensitivity than people with dementia.” This sentence would be more helpful with specific reasons: e.g., subtle deficits that may not be detected with less sensitive measures, wider range of premorbid abilities than populations the tests were normed on, wider range of age ranges. More detail on this would help the battery feel more specific for post COVID.

- “In addition, conducting a cognitive assessment in uncontrolled settings could bias the results obtained”. More detail is needed here. Is this referring to telephone or internet assessments?

- The introduction felt very long in the second half and information page 4 paragraph 2 and 3 seemed like they could fit better in the methods section.

- The introduction could be stronger if more sections focused on the specifics of why this is the appropriate next step for post COVID condition, versus just in general. This would be really helpful in page 3, paragraph 4, and page 4 paragraph 1.

o Subtle attentional/processing speed deficits may be detected easier with computerized testing. Why is this relevant to long COVID? Some of the literature on deficits being subtle in this area may be really helpful and justify this measure.

Methods:

- Was this sample recruited from patients? Or advertised through the Department of Neurology? As neurological disorders were exclusionary, were these patients who reported neurological symptoms but did not meet criteria? Were the 508 participants meeting criteria and enrolled a small percentage of patients seen in the clinic? I feel like this would be a different sample than a community healthy control and more detail would be helpful (in the appendix if not space in the main paper).

- Did neurological disorder include minor neurocognitive disorder or mild cognitive impairment?

- You use cognitive complaints in this section and cognitive symptoms in the introduction. Using the same language for self-reported symptoms and objective performance based impairments throughout the article would be helpful.

- Statistical methods felt very clear to me and detailed enough to be replicated.

Results:

- On page 8, you talk about hospital and ICU admission. It would be helpful to have more detail in the methods section about how you determined this and where you obtained the information.

Discussion:

- The whole paper and discussion would benefit from making the aims of the paper a bit more clear. Is the primary aim to explore the factor structure in a combined sample of PCC and HC with a secondary aim of seeing the factor structure in the PCC group? Or is the primary aim exploring the factor structure in the combined sample and then using the computer testing to examine PCC specific deficits? I think this would be more clear to readers if you used more explicit language about the primary and secondary aims.

- On page 10, you mention additional adjustments in the WAF divided in RT reaction time. I did not see this mentioned in the results section.

- For previous studies highlighting attentional deficits, it would be helpful to have all domains compared, not just attention. Also, it would be helpful to note if they only measured attention. This is touched on a bit with memory but more detail would be helpful.

- Was COVID infection severity similar in the articles you cite with similar findings?

- For page 10, you state visual memory may be less sensitive than verbal episodical memory tests. It would be helpful to have supporting literature from other test batteries or other populations to bolster this claim.

- Were there performance validity tests included? If not, this should be listed as a limitation in the discussion section and something that could potentially be added to future studies.

6. PLOS authors have the option to publish the peer review history of their article (what does this mean? ). If published, this will include your full peer review and any attached files.

**Do you want your identity to be public for this peer review?** For information about this choice, including consent withdrawal, please see our Privacy Policy .

Reviewer #1: No

Reviewer #2: **Yes: ** Dr. Kalliopi Megari

Reviewer #3: No

---

## [Author Response · Author response to Decision Letter 1]

14 Jan 2025

RESPONSES TO THE EDITOR AND REVIEWERS’ COMMENTS

Dear Editor and Reviewers,

Attached you will find the revised version of our manuscript “Computerized neuropsychological assessment in post-COVID condition” (PONE-D-24-46762).

We thank the editor and reviewers for their interesting observations and contributions to our article, and the opportunity to submit a revised version of the paper. We address each of their commentaries below:

Editor:

Thank you for addressing the reviewers' comments in a revised version while focusing on reviewer 3 and reviewer 1 comments. In addition, please provide greater details regarding how PCC (WHO) definition was exactly applied in the PCC sample. That is for each individual how the WHO was applied on each relevant criterion of the definition and using which tools.

Response: The diagnosis of PCC followed the recommendations and definition provided by WHO (Soriano et al., 2022). It is based on the examination and observation of the patients, paying attention to the evolution and the symptoms. Currently, there are no validated diagnostic tests for PCC.

We only included patients who met the criteria according to indications, excluding those with any neurological disorders involving cognitive impairment or other systemic or psychiatric conditions that could potentially impact the findings. All participants in our study also underwent neuroimaging tests (MRI in all cases) to exclude other causes. Additionally, CSF biomarkers of neurodegeneration and FDG-PET were conducted whenever there was even minimal suspicion of a neurodegenerative process.

We have completed the Methods section specifying these aspects.

Was fatigue measured at any stage? And other key post-viral syndromes? Was this also completed in the health controls? How comorbid causes were excluded (or not) as explanatory (or not) of the current post-viral symptoms as required by the WHO definition? Was a clinical examination conducted and if so by whom and using which instruments?. This additional information is needed to establish the robustness of the PCC definition in the current study.

Response:

Fatigue was assessed at the time of assessment using the Modified Fatigue Impact Scale (MFIS).

Post-viral syndromes were ruled out taking into account the patient's medical history. Patients with a previous post-viral syndrome were specifically excluded.

For patient selection, both physical and neurological examinations were performed. The psychical examination also facilitated the assessment of potential organ damage. Additionally, complementary tests such as blood tests, X-rays, and brain MRI scans were conducted when necessary to exclude other underlying causes. In addition, to exclude other potential causes (such as psychiatric disorders, post-viral syndromes) all patients were interviewed, and the information was cross-checked with the hospital and regional medical chart.

Following the Editor’ suggestions, we have completed the Methods section accordingly.

Furthermore, please provide additional information regarding acute infection including, how long after acute COVID infection, were the participants evaluated? What types of SARS-CoV-2 variants were most prevalent at the time of this study (by participants of these data is available), how many were vaccinated, how many have received antiretrovirals? Include these as limitations if these data were not available and discuss how this may have impacted the results.

Response:

Patients were evaluated 17.59 ± 8.86 months after infection.

The specific type of SARS-CoV-2 variants was not available in our study, but we had information about the waves based on the date of infection of the patients. Based on data from the Spanish Ministry of Health (2024), the first wave appeared between March 2020 and May 2020 (first strain predominated); the second wave occurred between August and November 2020 with the alpha variant (the B.1.1.7), the third wave occurred between December 2020 and July 2021 with the delta variant (the B.1.617.2), and the last wave, from December 2021 to the present, with the predominance of the omicron variant (B.1.1.529). (These data come from the Spanish Ministry of Health (2024); Spanish Ministry of Health, “SARS-CoV-2 disease in Spain”. Updated July 11, 2024). (Data from the Spanish Ministry of Health, 2024).

Based on the first date of symptom onset of the acute infection was used to determinate the waves of infection: one hundred forty patients (61.7%) belonged to the first wave (March-May 2020), thirty-eight (16.7%) to the second wave (alpha variant, August-November 2020), seventeen (7.5%) to the third (delta, December 2020-July 2021), and thirty-two (14%) to the fourth or subsequent waves. We have completed the manuscript with this information.

Regarding the vaccination status: 221 (96.4%) patients were not vaccinated on the first date of infection. At the time of cognitive assessment, 156 (68.7%) patients were vaccinated.

Treatment received with antiretrovirals during the acute phase was not included in the analysis because the information was not available for all the patients. However, in our sample, only 23.9% of patients were hospitalized. Considering that antiretrovirals were mainly administered to hospitalized patients during the first wave, we believe it is unlikely to affect the results in our sample, given that this represents a small proportion of patients. We have completed the limitations section as suggested.

Could you also explain why the current study participants were recruited from a Neurology tertiary service? What are the consequences of the current samples’ ascertainment? These data are needed to better establish the representativeness of the sample in comparison to the Spanish general population at the same time of the pandemic. Please include references to relevant Spanish general population data.

Response: Patients were recruited from our hospital and we included patients who consulted due to persistent cognitive issues and other symptoms, such as fatigue, after COVID-19. Madrid, and especially our area, has a healthcare system providing easy access to Neurology. In this regard, although we are a tertiary center providing specialized care, we have direct access from primary care. In fact, we have a direct access from primary care to the neurologists caring for patients with post-COVID, avoiding the bias of selecting the more severe cases. Additionally, the free choice of hospital in Madrid also allows to the patient to select a specific hospital and medical professionals. Patients made their choices based on various factors, such as knowledge of other patients who had been treated, recommendations, specialization, and other considerations. (Matías-Guiu et al., 2015). This allowed us to reach a larger sample size.

The data included in our study are in accordance with previously published studies in the Spanish population. PCC is more prevalent in women, aged 30-55 years and affects patients with both mild and severe forms of the disease (Ledo et al., 2021; López-Sampalo et al., 2022). Moreover, our sample is consistent with other studies carried out in the Spanish population with PCC regarding cognitive symptoms, in demographics and clinical terms (Ariza et al., 2022; García-Sánchez et al., 2022; Serrano del Pueblo et al., 2024).

Reviewer #1:

1-Thank you for allowing me to review this interesting and clinically-needed research Throughout the paper, the term Post COVID condition (PCC) is used. It is unclear if this refers only to persistent cognitive symptoms. Please include how you are defining the term.

Can you please describe why 6 months was chosen for persistent symptoms, as opposed to 3 months which is the definition of PCC.

Response:

Thanks for all the positive comments and suggestions to improve the paper. Post-COVID Condition (PCC) refers to the condition defined by the WHO (WHO, 2021). This condition occurs in individuals with a history of SARS-CoV-2 infection approximately 3 months from the onset, and symptoms lasting for at least 2 months, cannot be attributed to alternative diagnoses (Soriano et al., 2022). The diagnosis is based on the examination and observation of the patients, paying attention to the evolution and the symptoms. Currently, there are no validated diagnostic tests for PCC.

Patients were selected with a 6-month criterion. The 6-month criterion was used to separate possible symptoms that may occur after the acute or subacute phase of the disease (inflammation) and persistent symptoms. This criterion was established to be able to separate those possible cases where there may be a spontaneous recovery and to recognize those patients in whom the symptoms have become chronic (Correa & Vallespín, 2022). In other similar pathologies such as Chronic Fatigue Syndrome (CFS), the 6-month criterion is also used to define this condition as chronic (Fernández et al., 2009). We have completed the manuscript with this information.

2-Could the HC group include other forms of Long COVID? For instance, for people who report post-covid persistent changes in smell perception, what group would they go into?

Response: The recruitment of control subjects was carried out through the Neurology Department by reaching out to patients’ relatives, caregivers, and friends, as well as in collaboration with neighborhood associations in Madrid and nearby areas. Participants were asked about their COVID-19 infection history, and a questionnaire on potentially associated symptoms was administered to confirm that these individuals did not experience any symptoms following the infection. We have completed the Methods section accordingly.

3-Methods: Can more detail be provided on the administration of the measures. Were they done in person or remotely? What was the role of the administering neurologists? Did they read instructions and clarify? Can discussion of whether the battery could be used remotely be added? Has the Vienna Test System been validated? Has it been shown to correlate with other neuropsychological measures?

Response: The computerized battery was administered in person at our hospital by trained neuropsychologists. It was conducted in two-day sessions, following the battery guidelines. Each session lasted approximately 80 minutes. Before testing, the neuropsychologist provided a summary of the tests’ aims, duration, and content of testing. Participants were also informed about how data would be used (e.g. anonymization or documentation). Each participant was required to sign the informed consent form and was reminded to use their glasses and hearing aids, if necessary. The instructions and procedures are standardized, and during test administration, the administrators were instructed not to interfere with the instructions, as participants were expected to work independently (ideally without assistance). However, the neuropsychologist could provide assistance to resolve any issues or doubts. Another important aspect involved observing and recording the participant's conduct and behavior during the test.

We have included more details in the Methods section.

Regarding the normative data, the tests battery has several standardization studies in different countries and different samples (adults, children’s and different professional samples) (Aschenbrenner et al., 2012; Sturm, 2018). More details are explained below:

For the COGBAT Index: the norm sample for the standard form was collected (2012-2013) from the research laboratory at SCHUHFRIED GmbH and in cooperation with the teaching and research activities at the University of Vienna. Norming process involved a comprehensive sample of normal people that in terms of gender and age was representative of the general population of Germany, Austria, and Switzerland the data relates to an Austrian sample of 419 individuals from the normal population aged between 16 and 80. The sample consists of 184 (44%) men and 235 (56%) women. Participants were only included if they could state that they had not previously suffered from any serious neurological or psychiatric illness. Another representative norm sample was collected in Italy. The representative Italian norm sample was collected at multiple locations in Italy on different occasions. The samples of all tests correspond to the age and gender distributions gained from the demographic data from Italy from 2012 (Aschenbrenner et al., 2012).

Regarding WAF battery: the data of the representative norm sample of 306 individuals was gathered in 2016-2017 in the Test and Research Center of SCHUHFRIED GmbH, using a quota plan stratified according to gender, age and education. The expected age and gender distribution was obtained from demographic data for Germany, Austria, and Switzerland for 2016 (European Commission, 2016). The norm sample consists of 157 (51%) women and 149 (49%) men aged between 14 and 88 (Sturm, 2018). Neglect add-on subtest of the WAF and FLEI add-on were normed separately. The norm data was collected in 2011 in the research laboratory of SCHUHFRIED GmbH. The sample involves 329 respondents taken from the normal population. The sample consists of 151 (46 %) men and 178 (54 %) women. The respondents range in age from 16 to 83 (Beblo et al., 2012).

Other independent tests also have normalization studies. For example, DT: The data of the representative norm sample of 759 individuals was gathered in 1999 - 2017 in the Test & Research Center of SCHUHFRIED GmbH, using a quota sample plan stratified according to gender, age, and education in multiple collection waves. The data of the latest data collection wave was collected from 2016 to 2017 (Schuhfried, 2020); COG: the data of the representative norm sample of 662 individuals was gathered in 2005 - 2015 in the Test & Research Center of SCHUHFRIED GmbH. The data for the norming were collected in two data collection samples between 2005-2007 and in 2015 (Schuhfried, 2013); RT: the representative norm sample of 673 individuals was gathered from 2014 to 2016 at several locations in Italy using a quota sample plan stratified according to gender and age. The expected age and gender distribution was obtained based on demographic data for Italy for 2011 (Istituto Nazionale di Statistica, 2011). The norm sample consists of 311 (46%) men and 362 (54%) women aged between 16 and 94. There are studies in different countries (Slovakia, Brazil, Poland, Netherlands, France…) and with different samples such as professional drivers, Traffic psychological clients, athletes, machine operators (Schuhfried, 1996).

Correlation with other measures:

VTS has demonstrated validity and reliability comparable to traditional neuropsychological measures. Many tests are designed to assess functions, such as attention, memory, processing speed, executive function, and motor skills, and are often compared with other well-established neuropsychological tests.

Some examples of this: TMT-L was compared with the performance of the paper-and- pencil version from Reitan (1992) to test the construct validity of the TMT-L and significant correlations were found for all pairs of variables at the 0.01 level (Rodewald et al., 2012); FGT has shown correlations with a number of other tests, including tests of attention and various executive functions, have been investigated; the results provide evidence of the test’s construct validity (Vetter, J.; Aschenbrenner, S. & Weisbrod, 2018); DT: had shown correlation analyses with a number of further tests from the Cattell-Horn- Carroll model (CHC model), which are often applied internationally in the framework of traffic psychological investigations, prove construct validity of the test’s characteristic interpretations. Furthermore, there is a large number of studies in the areas of traffic psychology, safety assessment, sports and clinical (NEURO) psychology which prove the criterion validity of DT in different application contexts (Schuhfried, 2020); COG: Based on the CHC model (Schneider & McGrew, 2018) the convergent and discriminant validity for the COG was tested with a number of further tests (LVT, VISGED, ATAVT, Corsi, DT, RT) (Schuhfried, 2013); RT: RT has the highest correlation with the COG and DT tests as can be expected from the no

---

## [Decision Letter · Decision Letter 1]

19 Mar 2025

Computerized neuropsychological assessment in post-COVID condition

PONE-D-24-46762R1

Dear Dr. Matias-Guiu,

We’re pleased to inform you that your manuscript has been judged scientifically suitable for publication and will be formally accepted for publication once it meets all outstanding technical requirements.

Kind regards,

Lucette A Cysique, PhD

Academic Editor

PLOS ONE

Additional Editor Comments (optional):

Thank you for addressing the comments of the reviewers and that of the editor.

Reviewers' comments:

Reviewer's Responses to Questions

**Comments to the Author**

1. If the authors have adequately addressed your comments raised in a previous round of review and you feel that this manuscript is now acceptable for publication, you may indicate that here to bypass the “Comments to the Author” section, enter your conflict of interest statement in the “Confidential to Editor” section, and submit your "Accept" recommendation.

Reviewer #1: All comments have been addressed

2. Is the manuscript technically sound, and do the data support the conclusions?

Reviewer #1: Yes

3. Has the statistical analysis been performed appropriately and rigorously? 

Reviewer #1: Yes

4. Have the authors made all data underlying the findings in their manuscript fully available?

Reviewer #1: Yes

5. Is the manuscript presented in an intelligible fashion and written in standard English?

Reviewer #1: Yes

6. Review Comments to the Author

Reviewer #1: Thank you for addressing each comment in a thorough and thoughtful way and for adding information into the manuscript to make it more understandable for the audience.

7. PLOS authors have the option to publish the peer review history of their article (what does this mean? ). If published, this will include your full peer review and any attached files.

**Do you want your identity to be public for this peer review?** For information about this choice, including consent withdrawal, please see our Privacy Policy .

Reviewer #1: **Yes: ** Amber Sousa

---

## [Editor Report · Acceptance letter]

PONE-D-24-46762R1

PLOS ONE

Dear Dr. Matias-Guiu,

I'm pleased to inform you that your manuscript has been deemed suitable for publication in PLOS ONE. Congratulations! Your manuscript is now being handed over to our production team.

Kind regards,

on behalf of

Dr. Lucette A Cysique

Academic Editor

PLOS ONE